

# Pharmacist-social worker interprofessional relations and education in mental health: a scoping review

Paul Boylan[1], Jamie Knisley[2], Brandt Wiskur[3], Jessica Nguyen[1], Kristine Lam[1], Jisoo Hong[1] and Joshua Caballero[4]

[1] College of Pharmacy, University of Oklahoma Health Sciences Center, Oklahoma City, Oklahoma, United States
[2] Total Dose, Edmond, Oklahoma, United States
[3] Office of the Vice Provost for Academic Affairs and Faculty Development, University of Oklahoma Health Sciences Center, Oklahoma City, Oklahoma, United States
[4] College of Pharmacy, University of Georgia, Athens, Georgia, United States

Corresponding author
Paul Boylan, paul-boylan@ouhsc.edu

## ABSTRACT

**Background:** One in eight patients is affected by a mental health condition, and interprofessional mental health teams collaborate to improve patient care. While pharmacists and social workers are recognized as mental health team members, there is a lack of literature describing interprofessional relations and education between these professions, especially as it pertains to mental health. The purpose of this review was to identify and characterize reports describing pharmacist-social worker interprofessional relations and education within mental health.

**Methodology:** To address this knowledge gap, this scoping review was conducted to collect and characterize reports published between January 1, 1960 and August 18, 2023 describing pharmacist-social worker interprofessional relations and education within the field of mental health. The Preferred Reporting Items for Systematic Reviews and Meta-Analyses extension for Scoping Reviews (PRISMA-ScR) guidelines were followed. Ovid MEDLINE, CINAHL, and Social Work Abstracts were searched using keywords "pharmacy student," "pharmacist," "social work student," "social worker," and "social work." Reports were included if they were published in English and interprofessional relations or education occurred directly between (student) pharmacists and social workers.

**Results:** Three hundred twenty records were identified and three records were included: one cross sectional study, one qualitative educational project, and one case report. Each record suggested positive patient and/or educational outcomes developing from pharmacist-social worker interprofessional relations and education. In clinical practice, pharmacist-social work teams identified mental health risk factors, reduced 30-day readmissions, and improved post-discharge telehealth care. In the classroom, a social worker improved pharmacy students' confidence assessing patient suicidal ideations.

**Conclusions:** This scoping review identified needs and areas for future research: pharmacist interprofessional education with Master of Social Work and Doctor of Social Work degree students, transitional care and mental health outcome measure reporting using evidence-based outcomes, and development of scholarly teaching projects utilizing higher-level educational frameworks beyond learner reactions.

# INTRODUCTION

Mental health is defined by the World Health Organization as a state of well-being that enables people to cope with life stressors, realize their abilities, learn and work effectively, and contribute to community (*World Health Organization, 2022a*). It is estimated that one in eight people worldwide are affected by a mental health condition, which can lead to significant distress, impaired function, and an increased risk of self-harm (*World Health Organization, 2022b*). Examples of common mental health conditions include depression, anxiety, and schizophrenia (*Kessler et al., 2009*; *World Health Organization, 2022b*). Serious mental health disorders significantly influence cognition, emotion, and behavior (*Kessler et al., 2009*; *World Health Organization, 2022b*). Every year, mental disorders have the potential to influence suicidal ideations and behaviors, with more than 700,000 suicide-attributable deaths or suicide attempts recorded worldwide (*World Health Organization, 2022b*).

According to the National Center for Health Statistics' 2013 Report on mental health access, an astonishingly low percentage of patients interacted with mental health professionals (*Decker & Lipton, 2015*). Among adults aged 18 to 64 years with private or state-sponsored insurance, less than 40% interacted with a mental health care provider in the past 12 months (*Decker & Lipton, 2015*). Among uninsured adults, less than 20% reported interacting with a mental health care provider within the same timeframe. (*Decker & Lipton, 2015*). It is worth noting that this survey question specifically mentioned psychiatrists, psychologists, psychiatric nurses, or clinical social workers as mental health professionals (*Decker & Lipton, 2015*). Pharmacists were not included on this list, despite the fact they are recognized as medication experts on interprofessional mental health teams and can obtain board certification in psychiatry pharmacy (*i.e.*, Board Certified Psychiatric Pharmacist (BCPP)) (*Dopheide et al., 2017*, *2022*; *Wenthur, Gallimore & Zorek, 2021*). Pharmacists commonly interact with prescribers (*e.g.*, psychiatrists, mid-level providers) and psychologists. However, social workers and pharmacists also spend time communicating to determine access to medications and environmental factors (*e.g.*, family support, living arrangements) which may impact adherence and health outcomes (*e.g.*, readmissions) (*Dopheide et al., 2017*; *Wenthur, Gallimore & Zorek, 2021*). Bentley and Walsh propose six roles for mental health social workers: collaborator, consultant, advocate, monitor, educator, and researcher (*Bentley & Walsh, 2006*; *Walsh, 2008*). These roles are similar to those endorsed by the Center for the Advancement of Pharmacy Education educational outcomes for pharmacists (*Medina et al., 2023*, *2013*).

Both pharmacy and social work educational accreditation standards require interprofessional education and practice in didactic and experiential training for clinical practice (*Accreditation Council for Pharmacy Education, 2016*; *Council on Social Work Education, 2022*). Leaders in both professions have strongly supported expansion of

interprofessional education beyond the required interactions with physicians or prescribers (*Brazeau, 2013*; *Oliver, 2013*; *Rubin et al., 2017*). Members of the American Association of Psychiatric Pharmacists (AAPP) recommend the topic of mental health and the importance of psychiatric interprofessional teams beginning in the first professional year of Doctor of Pharmacy (Pharm.D.) degree curricula (*Dopheide et al., 2017*). One scoping review characterizing interprofessional efforts in mental health crisis response systems found similar support for interprofessional collaborations, but acknowledged a lack of representation across professions, including pharmacist-social worker relations and education (*Winters, Magalhaes & Kinsella, 2015*). As early as 2011, medication therapy management was proposed as a way to foster pharmacist-social worker interprofessional relations and education; (*Rust & Davis, 2011*) however, medication therapy management has since evolved into comprehensive medication management (*American College of Clinical Pharmacy et al., 2015*). One report by MacDonnell and colleagues described an interprofessional exercise involving students including third-year pharmacy, second-year graduate social work, fourth-year nursing, second-year medicine, and second-year physical therapy tasked with identifying and managing a patient case simulating a domestic violence encounter (*MacDonnell et al., 2016*). Few, if any, follow-up reports have been published that describe solely pharmacist-social worker interprofessional relations and education within mental health. Two Cochrane systematic reviews suggest the benefits of interprofessional and multidisciplinary teams to improve outcomes for patients with mental health conditions (*Darker et al., 2015*; *Marin et al., 2017*). Mental health benefits gleaned by patients include reduced benzodiazepine dependence, reduced disability, improved pain scores, improved appointment attrition, and better transitions of care. These health outcomes may have been improved though motivational interviewing and patient counseling (*Deng et al., 2022*). Such methods require the identification of barriers to adherence which may include stigma and lack of resources which can more likely be identified by social workers while medication adverse events, proper administration, and efficacy are triaged by pharmacists (*i.e.*, prime questions) (*Cormier, Nurius & Osborn, 2016*; *U.S. Public Health Service and Indian Health Service, 1991*). Recent literature has placed a focus on providing precision counseling to optimize adherence and potentially impact health outcomes (*Caballero, Jacobs & Ownby, 2022*). Precision counseling is the ability to assess social stressors, lifestyle, environmental factors, and medication adverse events to impact adherence which dovetails into areas social workers and pharmacists address in their practices. However, these data and current body of literature have not specifically examined pharmacist-social worker interprofessional relations or education, (*Darker et al., 2015*; *Marin et al., 2017*) and merits review. Therefore, the interprofessional education, relations, and outcomes between social workers and pharmacists within mental health have not been fully elucidated. The purpose of this scoping review was to collect and characterize reports describing pharmacist-social worker interprofessional relations and education within mental health.

## SURVEY METHODOLOGY

### Review scope and question

This was a scoping review of the literature on interprofessional education and relations between pharmacists (including student pharmacists) and social workers (including social work students). This scoping review utilized the Preferred Reporting Items for Systematic Reviews and Meta-Analysis extension for scoping reviews (PRISMA-ScR) protocol (*Tricco et al., 2018*). This scoping review was not registered in the International Prospective Register of Systematic Reviews (PROSPERO) because scoping reviews are ineligible for PROSPERO registration. Scoping reviews follow similar methodology to systematic reviews but scoping reviews do not aim to answer a focused patient-intervention-comparison-outcome question. Instead, scoping reviews aim to collect research on a topic and identify gaps to inform future investigation (*Tricco et al., 2018*). The research question this scoping review aims to answer is: what reports describe and characterize interprofessional relations and education between pharmacists and social workers within mental health?

### Data sources and search

An electronic literature search was performed using Ovid MEDLINE, CINAHL, and Social Work Abstracts. Medical subject headings included: "pharmacist" or "student pharmacist" and "social work," "social worker," or "social work student." In order to collect a scoping sample of literature, "mental health" was not used as a keyword during the search to avoid excluding articles that were relevant to mental health but not indexed with the "mental health" medical subject heading. The abstracts and full-text reports were screened by two authors (JK, JN, KL, or JH and PB) for their relevance to mental health.

### Report selection

Inclusion and exclusion criteria are charted in Table 1. To clarify, MEDLINE medical subject headings define interprofessional relations as "reciprocal interactions between two or more professionals," and interprofessional education as "education encouraging healthcare professionals to learn each other's roles and responsibilities to provide patient-centered care" (*National Library of Medicine, 2022*). Notably, reports describing pharmacist and social worker interactions alongside other healthcare professions (*e.g.*, medicine, nursing, clinical psychology) were excluded so as to focus results specifically on pharmacist-social worker interactions and education, and to minimize collider bias occurring from interprofessional interactions and education from other healthcare professions (*Holmberg & Andersen, 2022*).

### Data charting and synthesis

Results from the searches were exported as a Research Information Systems file and imported into Covidence (Covidence, Melbourne, VI, Australia) for abstract screening, full-text review and assessment, and data abstraction. Abstracts were independently screened for inclusion by two authors (JK, JN, KL, or JH and PB) and screening conflicts were resolved through in-person discussion and agreement. Full text reports were

**Table 1 Scoping review inclusion and exclusion criteria.**

| Inclusion criteria | Exclusion criteria |
|---|---|
| • (Pharmacist OR student pharmacist) AND (social worker OR social worker student OR social work) AND interprofessional education | • One-on-one interviews of pharmacists-social workers (*i.e.*, missing direct interprofessional education or relations between pharmacists and social workers) |
| • (Pharmacist OR student pharmacist) AND (social worker OR social worker student OR social work) AND interprofessional relations | • Interprofessional education or relations reports including health professions beyond pharmacists and social workers |
| • Randomized controlled trials, nonrandomized trials, observational studies, descriptive reports, and case series or case reports | • Systematic reviews with or without meta-analyses, newsletters, editorials, theses, commentaries, and citations without abstracts |
| • Quantitative, qualitative, or mixed-methods | • Published in a language other than English |
| • Published between 1/1/1960 and 8/18/2023 | |
| • Published in English language | |

independently screened for inclusion by two authors (JK, JN, KL, or JH and PB) and screening conflicts were resolved through in-person discussion and agreement.

Two authors (JK and PB) independently completed data extraction and conflicts in charting were resolved through in-person discussion and agreement. Data were extracted using an existing template in Covidence. Reports describing interprofessional education were evaluated using Kirkpatrick's educational frameworks (*Haines & Seybert, 2021*; *Kirkpatrick & Kirkpatrick, 2006*). A bias assessment of the reports was not performed as it is not standard of practice for scoping reviews (*Tricco et al., 2018*). Authors noted if any component of the reports assessed suicide prevention, treatment, or education.

## RESULTS

Figure 1 depicts the PRISMA flow diagram for screening and full text reports meeting inclusion criteria (*Page et al., 2021*). Three articles were included in this scoping review and are charted in Table 2 (*Gil et al., 2013*; *Reaume, 2023*; *Witry et al., 2019*).

One retrospective cross-sectional report described pharmacist-social worker interprofessional relations in transitional care at an academic medical center in the United States (*Gil et al., 2013*). Patients identified for the pharmacist-social worker transitional care program were eligible if they possessed at least one medication or psychosocial risk-factor from one of three categories: high-risk medications (*i.e.*, opioids or psychotropics), clinical risk factors (*i.e.*, mental health conditions including depression, substance use, or dementia), and psychosocial risk factors (*Gil et al., 2013*). Pharmacist-social worker interprofessional relations and collaborative care occurred *via* four program components: daily inpatient rounds; medication history, assessment, and education; discharge medication review; and post-discharge telehealth follow-up (*Gil et al., 2013*). The pharmacist-social worker team identified 10% of patients possessed a risk factor for mental health (*Gil et al., 2013*). Compared to the study site's standard care (*i.e.*, physician or nurse-led transitional care), patients in the pharmacist-social worker cohort experienced significantly fewer 30-day readmissions *vs.* standard care (10% *vs.* 30%, $p = 0.012$) (*Gil et al., 2013*).

One retrospective qualitative report described first-year student pharmacists' interprofessional education interacting with a social worker in the didactic setting
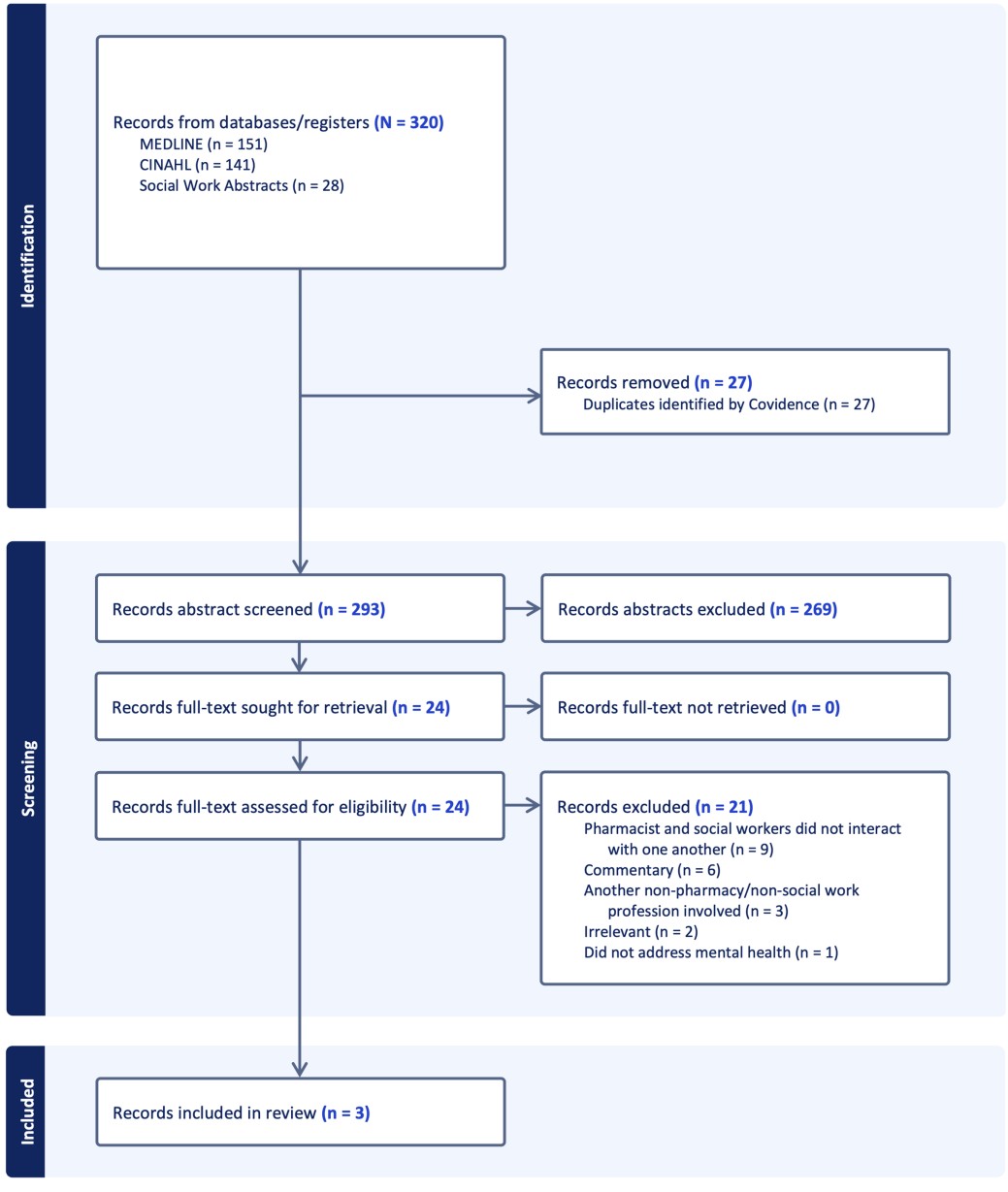

**Figure 1 PRISMA flow diagram of record abstract screening, full-text assessment, and article extraction.**

(*Witry et al., 2019*). A social work faculty delivered a 50-min session on suicide prevention which included introductory information on suicide and suicide awareness, communication and practical strategies to assess patients for suicidal ideation, community resource navigation, and suicide case scenarios with a question-and-answer session (*Witry et al., 2019*). Following the session, students answered multiple choice examination items on the content and were asked to complete a voluntary survey (*Witry et al., 2019*). Students' academic performance on the examination questions was unreported. However, the survey response rate was approximately 77% (83 out of 108 eligible students completed the survey). Sixty-six percent ($n = 55$) of respondents indicated confidence in their ability

**Table 2  Reports describing pharmacist and social worker interprofessional relations and education in mental health.**

| Study (year) | Location | Study design | Length of intervention | Study subjects | Intervention | Was suicide prevention, treatment, or education assessed? | Study outcomes | Limitations |
|---|---|---|---|---|---|---|---|---|
| Gil et al. (2013) | US | Cross-sectional | Multi-day | English-speaking adults >18 years with one or more high-risk readmission characteristics admitted to a medical-surgical unit at an academic medical center | Pharmacist-MSW social worker provided transitional care to admitted inpatients and provided telehealth post-discharge | No | Reduced 30-day readmissions in pharmacist-social worker cohort $vs$ standard care (10% $vs$ 30%, $p < 0.05$) | Single-center and single medical-surgical unit location; power calculation not performed; primary outcome of 30-day readmission rate is prone to confounding; findings may lack generalizability to non-English-speaking populations |
| Reaume (2023) | Canada | Case report | 30 weeks | 61-year-old male with chronic back pain and depression | Pharmacist optimized antidepressant pharmacotherapy and helped patient reduce basal opioid requirements concurrent to social worker-led cognitive behavioral therapy to reorient patients' maladaptations to pain his emotional distress | No | By 30 weeks, the patient's morphine equivalent dose was reduced by almost 50% and pain severity and pain interference by almost 40% Patient-reported outcome scores improved on questionnaires assessing depression, anxiety, stress, neuropathy, phobia, and pain | Case report of one subject; case followed for up to 30 weeks and long-term outcomes >1 yr were unreported; pain may be a subjective outcome measure; mental health domains including depression, anxiety, and stress were assessed individually rather than a global mental health symptom score |
| Witry et al. (2019) | US | Qualitative | Single day | 108 P1s enrolled in a required health services course, instructed by 1 MSW social worker faculty | 50-min session on suicide prevention delivered by MSW social worker faculty | Yes | 83 students (76.9% response rate) responded to post-session survey 83 students (76.9% response rate) answered two open-ended questions about concepts learned and potential topics for additional learning | Single-center and single day intervention; participant response rate <80%; inductive and deductive qualitative methods were undescribed |

**Note:**
MSW, Master of Social Work degree; P1, first professional year Doctor of Pharmacy student; US, United States; yr, year(s).

**Table 3 Assessment of studies using Kirkpatrick's educational frameworks.**

| Domain | Report information |
|---|---|
| Study (year) | *Witry et al. (2019)* |
| Educational methods | Didactic and practice-based |
| Educational environment | Classroom |
| Number of training sites | One |
| Kirkpatrick's educational frameworks achieved | Level 1: reactions<br>Level 2: knowledge, attitudes, or skills |
| Level 1 outcomes | |
| Positive reactions | Respondents reporting suicide topic was relevant ($n = 83$, 100%): 31 (37%) extremely relevant, 36 (43%) very relevant, and 13 (16%) somewhat relevant<br>Respondents reporting confidence in asking about suicide ($n = 55$, 66%): very confident ($n = 9$, 11%) and somewhat confident ($n = 46$, 55%)<br>Respondents reporting increases in confidence asking about suicidal ideation ($n = 76$, 91%): greatly increased ($n = 15$, 18%) and somewhat increased ($n = 61$, 73%) |
| Neutral reactions | Respondents reporting the session did not change their confidence asking someone about suicidal ideation ($n = 3$, 4%) |
| Negative reactions | Respondents reporting the suicide topic was not at all relevant: ($n = 0$)<br>Respondents reporting confidence in asking about suicide: not at all confident ($n = 22$, 27%)<br>Respondents reporting decreased confidence asking about suicidal ideation: somewhat decreased ($n = 0$) and greatly decreased ($n = 1$, 1%) |
| Level 2 outcomes | |
| Declarative knowledge of learning | Students' self-reported most important content learned: directness ($n = 28$, 33.7%), warning signs ($n = 12$, 14.4%), confidence ($n = 11$, 13.2%), making a difference ($n = 9$, 10.8%), pharmacy context ($n = 8$, 9.6%), awareness and responsibility ($n = 7$, 8.4%), and resources ($n = 5$, 6%) |

**Note:**
Data reported as: frequency (percent).

to ask a patient if they were contemplating suicide (*Witry et al., 2019*). Ninety-one percent ($n = 76$) of respondents indicated the session increased their confidence in their ability to ask a patient if they were contemplating suicide (*Witry et al., 2019*). One open-ended question in the survey asked students to state the most important concept learned from the session. Students' most common responses included directness ($n = 28$, 33.7%), warning signs ($n = 12$, 14.4%), confidence ($n = 11$, 13.2%), making a difference ($n = 9$, 10.8%), pharmacy context ($n = 8$, 9.6%), awareness and responsibility ($n = 7$, 8.4%), and resources ($n = 5$, 6%) (*Witry et al., 2019*).

One case report of a 61-year-old Canadian Male with chronic low back pain and a past medical history including depression received interprofessional care from a pharmacist-social worker team over 30 weeks (*Reaume, 2023*). The pharmacist optimized the patient's opioid and antidepressant pharmacotherapy and collaborated with the social worker to address the patient's emotional and affective drivers of pain. The pharmacist and social worker individually and collaboratively cared for the patient over 30 weeks to reduce opioid dependence and improve patient-reported outcome measures (*i.e.*, questionnaire scores) in pain, depression, anxiety, stress, neuropathy, and phobia (*Reaume, 2023*).

Table 3 charts the results of the retrospective qualitative study using Kirkpatrick's educational frameworks (*Kirkpatrick & Kirkpatrick, 2006*; *Witry et al., 2019*). Results from

the report achieved two lower-level education frameworks through assessing learner perceptions and knowledge (*Kirkpatrick & Kirkpatrick, 2006*; *Witry et al., 2019*).

## DISCUSSION

The purpose of this scoping review was to collect and characterize reports describing pharmacist-social worker interprofessional relations and education within mental health. Three studies reported positive outcomes (*Gil et al., 2013*; *Reaume, 2023*; *Witry et al., 2019*). One report suggested the pharmacist-social worker collaboration achieved a 20% absolute risk reduction in preventable 30-day readmissions (*Gil et al., 2013*).

One qualitative report posited increased abilities and confidence for student pharmacists to discuss suicide with patients after a 50-min interactive session with a social worker (*Witry et al., 2019*). A case report of an adult with chronic low back pain and depression benefitted from the clinical collaborations attributable to pharmacist and social worker interprofessional relations (*Reaume, 2023*).

Results from a national survey revealed that almost 99% of clinical social workers practicing in mental health encounter patients taking at least one medication to treat a mental health condition (*Bentley, Walsh & Farmer, 2005*). More than 50% of respondents reported assisting patients with weighing the risks and benefits of taking medications for mental health (*Bentley, Walsh & Farmer, 2005*). However, 32% of these respondents indicated they felt incapable of helping patients make medication-related decisions (*Bentley, Walsh & Farmer, 2005*). Between 85% to 95% of respondents felt medication management should be triaged to physicians (*Bentley, Walsh & Farmer, 2005*). These findings underscore the need for efforts to improve social workers' familiarity with mental health medications in clinical practice, ability to educate patients on the therapeutic benefits and potential harms associated with these medications, and awareness of the expertise of mental health pharmacists. Mental health pharmacists are perfectly positioned as the medication experts on the mental health team to collaborate with social workers in clinical practice and provide corresponding pharmacotherapeutic education (*Dopheide et al., 2017*). Social work students and clinical social workers can equally reciprocate by collaborating with pharmacists concerning nonpharmacologic therapy and sociobehavioral interventions in mental health. The reports identified in this scoping review demonstrated improved patient care outcomes and successful interprofessional education through pharmacist-social worker interactions (*Gil et al., 2013*; *Reaume, 2023*; *Witry et al., 2019*). Further studies should be conducted that address social workers' knowledge of mental health medications and assess their comfort caring for patients taking these medications. Conversely, future studies may also evaluate the knowledge pharmacists and student pharmacists obtain from social workers to optimize their medication counseling and interaction with patients. For example, pharmacists can focus on medication-related issues while social workers can identify additional risk factors for non-adherence (*e.g.*, transportation, insurance, living situation) and collaboratively engage in resolving such barriers.

This scoping review was conducted to gather and analyze existing literature on the interprofessional relationships and educational interactions between pharmacists and

social workers in the context of mental health care. The review successfully identified three studies that demonstrated positive patient and academic outcomes from such collaborations. Specifically, *Gil et al. (2013)* reported a significant outcome where the collaboration between pharmacists and social workers was associated with a 20% absolute reduction in the risk of preventable 30-day hospital readmissions. Additionally, a qualitative study by *Witry et al. (2019)* revealed that student pharmacists exhibited enhanced abilities and confidence in discussing suicide with patients following a 50-min interactive session led by a social worker. Furthermore, *Reaume (2023)* presented a case report illustrating the clinical benefits reaped by an adult patient suffering from chronic low back pain and depression, which were directly attributed to the collaborative efforts between pharmacists and social workers. We hypothesize that the positive outcomes of these studies may be related to the focus of these professions with pharmacists concentrating on medication-related factors while social workers are focused on environmental/social stressors. Pharmacists are traditionally guided by the three prime questions focused on the purpose of the medication, proper use/technique, and perceived benefits/side effects of the medication (*U.S. Public Health Service and Indian Health Service, 1991*). Social workers' counseling is concentrated on environmental factors such as living situation, support systems, challenges/concerns, coping strategies, and education/ employment (*Cormier, Nurius & Osborn, 2016*). As such, the collaborative efforts within both fields may be maximizing outcomes as it holistically addresses several factors impacting adherence. However, this hypothesis merits further study and investigation in mental health. Collectively, these findings may underscore the potential of interprofessional relationships in improving mental health outcomes and represent progressive findings beyond those previously published in systematic reviews (*Darker et al., 2015*; *Marin et al., 2017*).

Two reports in this scoping review included student pharmacists and pharmacists interacting with clinical social workers who possessed a masters degree (Master of Social Work (MSW)) (*Gil et al., 2013*; *Witry et al., 2019*). Social workers may choose to continue their education with either Doctor of Social Work (DSW) or Doctor of Philosophy (Ph.D.) in Social Work degrees (*Howard, 2016*). The creation of the DSW in the 1940s may be seen as analogous to the implementation of the Pharm.D. degree in the 2000s as the terminal professional degree for clinical practice and education (*Vlasses, 2010*). Our scoping review did not identify any reports describing DSW-Pharm.D. relations or education pertaining to mental health, or otherwise. Therefore, there may be potential for interprofessional collaboration between Doctors of Social Work and Pharmacy in areas outside mental health. Results from a national survey revealed that 88% of clinical social workers practicing in mental health possessed a MSW; DSW degree status was unreported (*Bentley, Walsh & Farmer, 2005*). It is recommended social workers' interprofessional education with pharmacists should be integrated in MSW curricula. This may allow for social workers to understand the role clinical pharmacists may play especially in mental health. Data suggest social work curriculum include additional training and knowledge regarding psychopharmacology and medication management (*Farmer, Bentley & Walsh, 2006*). Clinical psychiatric pharmacists may be positioned to provide this education given their

post-doctoral training and board certification available (*e.g.*, psychiatric pharmacy residency training, board certification in psychiatric pharmacy (BCPP)). Additionally, the role and impact of social workers should also be incorporated into pharmacy, especially their relationship to identify barriers to medication adherence are documented in other fields (*Tarfa, Pecanac & Shiyanbola, 2021*). Future studies should also improve reporting transparency in noting the terminal degree of social worker (*i.e.*, Bachelor of Social Work (BSW), MSW, DSW, PhD) and pharmacist (Bachelor in Pharmacy (B.Pharm.) or Pharm. D.) participants in the management of patients with mental health disorders.

The report by Gil and colleagues reported a 20% absolute decrease in 30-day readmissions after implementing pharmacist-social worker transitional care (*Gil et al., 2013*). Other randomized controlled trials and systematic reviews have similarly demonstrated pharmacists and social workers, albeit independent from one another, significantly reduce preventable 30-day readmissions (*Bronstein et al., 2015*; *Harris et al., 2022*). Thirty-day readmissions, although important to patients, professionals, and payors, are arguably a surrogate and composite outcome for patient-centered care and highly sensitive to countless confounding variables. Beyond 30-day readmissions, the Agency for Healthcare Research and Quality endorses discharge information communication and the provision of written discharge instructions as transitional care outcome measures (*Agency for Healthcare Research and Quality, 2016*). The National Inventory of Mental Health Quality measures includes over 300 evidence-based outcomes for documenting and researching mental health care (*Agency for Healthcare Research and Quality, 2012*). Therefore, reporting these additional transitional care and mental health outcome measures, and including racial and ethnic group information, was not present in the report by *Gil et al. (2013)* and should be included in future reports. Implementing these suggestions might enable a more precise quantitative assessment of the impact and facilitate an analysis of whether varying demographics yield comparable outcomes.

The findings of this scoping review suggest it may be beneficial for pharmacists and social workers in clinical practice to utilize the frameworks provided by these Agencies for tracking their interventions and evaluate the interprofessional communication between pharmacists and social workers. This approach could facilitate standardized reporting across various institutions and research studies. The report by Witry and colleagues reported one social worker-led educational session increased 76 student pharmacists' confidence in asking patients about suicide (*Witry et al., 2019*). This educational activity embraced mental health curricular recommendations posited by AAPP (*Dopheide et al., 2017*). Our scoping review determined Witry's activity achieved two of Kirkpatrick's educational frameworks: reactions and learning (*Kirkpatrick & Kirkpatrick, 2006*; *Witry et al., 2019*). Higher-level outcomes, specifically behavioral changes, have been suggested as the minimum reporting measures for scholarly teaching and learning (*Haines & Seybert, 2021*). Skills-based competencies are moderate-to-high level outcomes that are endorsed by social work and pharmacy education standards (*Accreditation Council for Pharmacy Education, 2016*; *Council on Social Work Education, 2022*). An example of one such activity was published by Washburn and colleagues wherein they developed and delivered an interprofessional mental health patient simulation to MSW students (*Washburn,*

*Bordnick & Rizzo, 2016*). Results showed MSW students significantly improved their diagnostic assessment skills and performance within an objective structured clinical exam (*Washburn, Bordnick & Rizzo, 2016*). Utilizing the AAPP member's curricular recommendations for pharmacy education in tandem with education accreditation standards from both pharmacy and social work may broadly help academicians and researchers in pharmacy and social work design educational activities that similarly deliver and measure higher-level outcomes including behavior change, performance, and learning habits (*Dopheide et al., 2017*; *Haines & Seybert, 2021*). Further reports describing and assessing high-level educational outcomes are needed.

## Limitations

This scoping review possesses limitations. We searched three databases–Ovid MEDLINE, CINAHL, and Social Work Abstracts–as suggested by PRISMA (*Tricco et al., 2018*). Additional databases, such as Scopus or Education Resources Information Center, may have captured additional reports that were not indexed in the three databases searched. Scoping reviews usually identify and extract between 20 and 30 articles (*Pham et al., 2014*) Fig. 1 reveals that we obtained and assessed 24 full-text articles that were relevant to our review question; however, nine studies were excluded because they were one-on-one semi-structured interviews or surveys lacking interpersonal pharmacist-social worker interactions as stipulated in our research question, and six studies were commentaries advocating for increased pharmacist-social worker collaboration. Although these articles did not meet the criteria to be included in our review, a follow-up review including qualitative projects and advocacy articles is warranted. Heterogenous terminology (often retired terms such as "anti-anxiety" instead of "anxiolytic," or "psychotropic" instead of "antidepressant") have been used in reports published in social work journals between the 1990s and early 2000s that pertain to mental health and medications (*Hughes & Cohen, 2010*; *Kensinger, 2007*; *Walsh, 2008*). Additional reports relevant to mental health may exist in the gray literature because of inconsistent terminology in report titles, abstracts, or keyword indices. Thus, relevant publications from the gray literature are unrepresented in this scoping review. We did not perform a bias assessment or a critical appraisal of the articles because it is not standard for scoping reviews (which differs from systematic reviews aiming to answer focused research questions) (*Tricco et al., 2018*). Completing a bias assessment may have added additional rigor to this review. Scoping reviews such as ours aim to map the current evidence base and propose further research endeavors. Additionally, although the reports in this scoping review published positive outcomes, there is potential for publication bias and there are opportunities for future reports to enhance methodology (*Gil et al., 2013*; *Reaume, 2023*; *Witry et al., 2019*). The nominal number of studies in our review may affirm an untapped need for additional investigations. This review's protocol was not registered in PROSPERO because scoping reviews are ineligible. Protocol registration in another database, such as Open Science Framework, may have been considered (*Lockwood & Tricco, 2020*); however, databases such as these lack the high-quality peer review associated with PROSPERO.

## CONCLUSIONS

This scoping review collected and characterized reports describing pharmacist-social worker interprofessional relations and education within mental health. The three studies identified in this scoping review highlight significant gaps in our understanding of interprofessional dynamics between pharmacists and social workers in mental health. To bridge these gaps, systematic and concerted efforts are needed to document and evaluate their collaborative practices and educational exchanges. These endeavors call for structured and collaborative frameworks that encourage both professions to engage in interprofessional academic and clinical experiences. Such approaches will enrich the educational experiences of pharmacists and social workers and enhance mental health care quality through a more cohesive and informed interprofessional partnership.

### Funding
The authors received no funding for this work.

### Competing Interests
Jamie Knisley is employed by Total Dose (Edmond, OK) as a post-graduate year 1 community pharmacy resident. The remaining authors declare that they have no competing interests.

### Author Contributions
- Paul Boylan conceived and designed the experiments, performed the experiments, analyzed the data, prepared figures and/or tables, authored or reviewed drafts of the article, and approved the final draft.
- Jamie Knisley conceived and designed the experiments, performed the experiments, analyzed the data, authored or reviewed drafts of the article, and approved the final draft.
- Brandt Wiskur analyzed the data, authored or reviewed drafts of the article, and approved the final draft.
- Jessica Nguyen performed the experiments, analyzed the data, authored or reviewed drafts of the article, and approved the final draft.
- Kristine Lam performed the experiments, analyzed the data, authored or reviewed drafts of the article, and approved the final draft.
- Jisoo Hong performed the experiments, analyzed the data, authored or reviewed drafts of the article, and approved the final draft.
- Joshua Caballero analyzed the data, authored or reviewed drafts of the article, and approved the final draft.

### Data Availability
   This is a literature review.

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
