# Peer review of "Pharmacist-social worker interprofessional relations and education in mental health: a scoping review"

_PeerJ, doi:10.7717/peerj.16977_

## Round 0.1 · original submission · Minor Revisions

Please address the following queries-

Keywords are missing from the abstract.

Could you mention the Scoping review guidelines that were followed in this review?

The RQ is missing. Show the framework used for RQ development.

Mention the IC/ EC in a tabular format.

Table 1 should include the limitations of individual publications.

·

Basic reporting

Paper is very clear and well written. The review is broad and of cross disciplinary interest and I feel it is within scope of the journal. Introduction is well done.

Experimental design

Well designed, narrow inclusion with excluding multidisciplinary education with other disciplines in addition to pharmacy and social but understand why it is necessary. Strict criteria for the review. May be beneficial to provide a sentence or two to the reader describing a scoping review in case they do not know the terminology.

Validity of the findings

Well done

Reviewer 2 ·

Basic reporting

The review meets the cross disciplinary interest of the journal.
The article reviewed the inter-professional education and relations between pharmacists and mental health education which is very important parameter for primary health care and is needed to be thoroughly studied.
However, I have following suggestions to further improve the article:
Abstract
- the objective of the review is required to be clearly stated (at line 30).
- the results should be presented with some specific findings at lines 41-42.
Introduction:
- The article adequately presented the Interprofessional roles of pharmacists and social workers however the importance of mental health education and how could it be translated into the patients' benefits are missing. you need to establish a motivation by proposing the this.
- the authors need to formulate a research question by which the review is guided (at lines 105-107).
- Pls give some details/insight on the research gap/ problem to which the review is focusing on. (lines 104-107)

Experimental design

The methodology is well-structured. However, I have following suggestion:
- As a scoping review gives importance on the far-reaching overview of literature and their findings representation, thus exclusion of Scopus database may not covered the findings comprehensively.
- Inclusion of expert opinion and presenting them to the audience might improve impact of the review.
- The exclusion, inclusion criteria should be described under separate sud-headings.

Validity of the findings

The authors need to provide a broad significance of its findings to address the goals of the review (at lines 309-318.
- The conclusion in the abstract mentioned that 'the article identified the areas and development of scholarly teaching projects utilizing higher-level educational frameworks beyond learner reactions (lines 43-47)' which was not reflected in the main discussion and conclusion of the study.
- Moreover, the discussion requires some detail supportive arguments and translation of the results to a the structural and developmental features in this area and are needed to be plotted
- Pls rephrase the conclusion so that it directs the readers to a structured and collaborative thinking for improving the interprofessional relationship between pharmacists and social workers.

Annotated reviews are not available for download in order to protect the identity of reviewers who chose to remain anonymous.

---

## Round 0.2 · accepted · Accept

Thanks for making the necessary changes.